# Vitamin D as a Biomarker of Ill-Health among the Over-50s: A Systematic Review of Cohort Studies

**DOI:** 10.3390/nu11102384

**Published:** 2019-10-06

**Authors:** Silvia Caristia, Nicoletta Filigheddu, Francesco Barone-Adesi, Andrea Sarro, Tommaso Testa, Corrado Magnani, Gianluca Aimaretti, Fabrizio Faggiano, Paolo Marzullo

**Affiliations:** 1Department of Translational Medicine, Università del Piemonte Orientale, 28100 Novara, Italy; silvia.caristia@uniupo.it (S.C.); nicoletta.filigheddu@uniupo.it (N.F.); francesco.baroneadesi@uniupo.it (F.B.-A.); andrea.sarro@uniupo.it (A.S.); tommaso.testa@uniupo.it (T.T.); corrado.magnani@uniupo.it (C.M.); gianluca.aimaretti@uniupo.it (G.A.); fabrizio.faggiano@uniupo.it (F.F.); 2Epidemiology Centre of Local Health Unit of Vercelli, 12100 Vercelli, Italy; 3I.R.C.C.S. Istituto Auxologico Italiano, Piancavallo, 28921 Verbania, Italy

**Keywords:** vitamin D, healthy ageing, cohorts

## Abstract

Background: The association between circulating levels of vitamin D and the incidence of chronic diseases is known. The identification of vitamin D as a biomarker of physiological/pathological ageing could contribute to expanding current knowledge of its involvement in healthy ageing. Methods: According to PRISMA guidelines, a systematic review was conducted on cohorts studying the role of 25OH-Vitamin D [25(OH)D] and 1,25(OH)_2_-Vitamin D [1,25(OH)_2_D] concentrations as biomarkers of healthy ageing. We consulted MedLine, Scopus, and Web of Science to search for studies on the association between vitamin D status in populations of originally healthy adults, and outcomes of longevity, illness, and physical and cognitive functionality. The quality of the studies was assessed using the Newcastle Ottawa scale. Results: Twenty cohorts from 24 articles were selected for this review. Inverse associations were found between low 25(OH)D levels and all-cause mortality, respiratory and cardiovascular events, as well as markers relating to hip and non-vertebral fractures. Associations between 1,25(OH)_2_D and healthy ageing outcomes gave similar results, although of lower clinical significance. Conclusions: This systematic review pinpoints peculiar aspects of vitamin D as a multidimensional predictor of ill health in the ageing process. Further well-designed controlled trials to investigate whether vitamin D supplement results in superior outcomes are warranted in the future.

## 1. Introduction

The promotion of healthy ageing is an essential objective in the strategy of making public health sustainable. Although health is not the only dimension of successful ageing, the reduction of the incidence and prevalence of diseases among older people can be considered to be one of the most relevant involved factors. For this purpose, the identification of biomarkers of healthy or successful ageing is an essential goal, to early detect unfavorable conditions and promote personalized prevention and treatment programs. Starting from general models of ageing [1], recent scientific research has studied the role of numerous biomarkers but has failed to identify which marker can “achieve the status of reliable predictor of biological ageing” [2]. Nevertheless, vitamin D has a potential dual role in the ageing process [3], since it can act both as a marker and as a possible therapeutic agent, and for these reasons it deserves, in our opinion, specific attention.

Vitamin D homeostasis is associated with biological and disease-related processes in the general population. A decrease of 25OH-Vitamin D [25(OH)D] concentrations has been observed in association with heart disorders, arterial hypertension, and atherosclerosis [4]. Receptors for the activated form of vitamin D have been identified in beta cells and immune cells, and vitamin D deficiency has been involved in the pathogenesis of both type 1 and 2 diabetes [5]. Moreover, vitamin D deficiency is predictive of cardiovascular events in patients with established cardiovascular disease (CVD) [6], while strong associations emerged in cohort studies and meta-analyses of prospective studies between vitamin D and cardiovascular mortality as well as all-cause mortality [7,8,9,10,11]. These relations possibly originate from the favorable actions of vitamin D on anti-inflammatory and endothelial functions [12], although the possibility of an inverse relationship cannot be excluded [3].

Vitamin D plays a role in human ageing. Older adults are more susceptible to develop clinical conditions relating to vitamin D deficiency due to decreased dermal vitamin D activation and age-related changes in lifestyle, adiposity, and physical exercise. Serum vitamin D levels in the range of deficiency can be documented in 2–30% of adults, a rate that rises to 75% in institutionalized older adults [13]. The effects of vitamin D on ageing also involves muscle health and musculoskeletal components of frailty in older people, such as poor physical performance, loss of muscle strength, sarcopenia, as well as risk of falls and fractures [14,15,16]. Moreover, vitamin D acts as a neurosteroid, and its association with the ageing brain entangles parameters of cognitive performance such as memory, orientation, and executive functions [17,18]. In vitro and in vivo data support evidence of a relationship between low 25(OH)D and several neurodegenerative conditions related to ageing, such as Alzheimer’s disease, Parkinson’s disease, and neurocognitive disorders [19]. 

Although large epidemiological evidence links vitamin D deficiency to multiple non-communicable systemic diseases that affect ageing, poor evidence of a direct causality exists, and hypovitaminosis D could simply act as a marker of illness [20]. However, trials and cohort studies on dietary vitamin D supplementation report evidence in support of the fact that maintenance of appropriate vitamin D levels in humans might prevent the gradual accretion of molecular and cellular damage associated with ageing and age-related diseases [21,22,23,24].

Research on factors promoting health in ageing has so far mostly focused on at-risk behavioral factors, which explains over one third of the global chronic disease burden [25]. Given the current rates of the substantial increase in life expectancy owing to medical and nonmedical factors [26], there is a need to reconcile such optimistic forecasts with the slower growth of healthy life years (HLY) [27]. A better comprehension of the current evidence and underlying mechanistic insights is of utmost importance to identify potential targets of action and propose interventions to promote healthy ageing through nutritional and treatment approaches. 

The aim of this study is to conduct a systematic review of cohort (or panel) studies on the role of 25(OH)D concentrations as a biomarker of healthy ageing, analyzing the association of 25(OH)D with the maintenance of health in middle-aged and older populations. 

## 2. Materials and Methods

A systematic review (SR) of longitudinal studies was conducted for the purpose of the study according to the PRISMA guideline (Preferred Reporting Items for Systematic Reviews and Meta-Analyses) [28,29]. The study protocol was drafted following PRISMA-P guidelines [30].

### 2.1. Eligibility Criteria

Studies included in this SR were selected according to the following criteria of eligibility:-Population: healthy people with a mean age over 50 years at study entry, who were untreated or not using pharmacotherapy; we also included populations at risk of biological/behavioral factors or showing abnormal biomarkers (except for hypertension), who were untreated, undiagnosed with specific diseases, and not taking dietary supplements or vitamins. We included articles with results relating to populations showing at baseline less than 25% of subjects with a prevalent diagnosis of a disease, or undergoing any pharmacological treatment, or taking vitamin D supplements. Articles were excluded if data on baseline health status of the cohort were missing;-Exposure: populations analyzed according to baseline levels of 25(OH)D or 1,25(OH)_2_D;-Controls: control groups enrolled from the same population of exposed people (for cohort studies) or as cases (for nested case-control in cohort studies);-Outcomes: indicators of the ten dimensions of the healthy ageing concept [31,32] (see Table 1);-Studies: the analysis included cohort studies, panel studies, and nested case-control in cohorts reporting analyses of associations between vitamin D measured at baseline and at least one outcome measured at follow-up, or analyses of validity/reliability of biomarker/s for the prediction of healthy ageing outcomes at follow-up by means of Receiver operating characteristic (ROC) curve and Area under the curve (AUC), optimal cut-off, test-retest, split-half and parallel form, internal concordance;-Setting: any type of setting except for public and private clinics for specialist care;-Time: papers published from January 2001 to March 2019;-Language: English, Italian, Spanish.

### 2.2. Search Strategy

Information sources used to search for literature evidence included databases from MedLine (PubMed), Scopus (Elsevier), and Web of Science. The search strategy was conducted by a reviewer expert in methodology and epidemiology science (SC) after a reassessment of biomedical, social, and psycho-social theories of ageing, further supported by the clinical and theoretical expertise of physicians (GA, PM), basic researcher (NF), and epidemiologist (FF) consulted for the purpose of the study.

The search in MedLine (PubMed) database was performed on 14 January 2019 using a string that included the MeSH Terms of aged, ageing, vitamin D and its synonymous, and healthy ageing combined with specific keywords. This first string was explorative and aimed to identify potential articles inherent to the PICOS search tool. Outcomes keywords and MeSH Terms regarded the dimensions of the healthy ageing concept. The search extended from January 2000 to January 2019. This first attempt returned 2194 records.

Then, the Scopus database was consulted on 5 March 2019, using unique strings for all types of healthy ageing outcomes in different combination with Boolean operators. The search was limited to journal articles in English, Italian, or Spanish published from 2000 to March 2019. A total of 209 records was collected. Subsequently, we searched in the Web of Science database on 6 March 2019 using the aforementioned general string regarding all type of healthy ageing outcomes and the same time and linguistic filters as previously detailed. From this database, 377 records were extracted.

Lastly, we further created nine strings for each type of outcome in the PubMed database on 28th March 2019, which returned a total of 1180 records. This search included longevity (219 records); cardiovascular, coronary, and cerebrovascular diseases (255 records); diabetes and metabolic syndrome (218 records); cancer and tumor (167 records); pulmonary and respiratory diseases (15 records); physical functionality (122 records); osteoporosis (126 records); cognitive functionality (31 records); healthy years (27 records). Filters were used to restrict the search to humans and time of publication from January 2000 to March 2019. All strategies are detailed in Appendix A Search strategy.

### 2.3. Study Records

Records emerging from search strings were reported in an Excel matrix, the same matrix was used to record data from the selection process related to the eligibility criteria of the SR.

After adjusting by duplicates, the selection process was conducted independently by two reviewers (TT, AS). In the first step, records were screened by title and abstracts, and finally by reading the full text of papers which passed the first step. Each article was registered in the matrix with a univocally assigned identification (id) number, presence or absence of criteria, and, consequently, if the paper passed each step. In the study eligibility screening phase, the process of full-text selection was recorded with mention of reasons for exclusion. Discordances among raters were discussed, in the wider team when necessary.

Data on publications included in the final step were reported in a separate matrix by one reviewer (TT), and the final matrix was controlled by the second reviewer (SC) to correct any errors in data entry.

### 2.4. Data Items

Each record was reported in a matrix with its univocal id, cohort name, number of participants, enrolment year, baseline year. For each record, the following information was collected: enrolment year, baseline year, number of participants, gender distribution at baseline, age (mean ± standard deviation (SD)) at baseline. Data on follow-up, such as mean frequency of follow-up, years from baseline to current follow-up, type of outcome, type of estimate, and results for each outcome were also included.

### 2.5. Outcomes Prioritization

Primary outcomes: markers of longevity, all-causes mortality, mortality for specific causes, diseases, ADL/IADL scales for physical functionality, and markers of cognitive functionality were included (Table 1).

Physical functionality in the healthy ageing concept is defined as the ability and autonomy in daily functional tasks normally measured by ADL and IADL scales [33]. However, clinical studies often use other indirect markers known to be associated with the impairment of daily functional tasks for the autonomy of older adults (i.e., risk of fracture, falls, physical performance tests, or hearing loss) [32].

Secondary outcomes: the indirect marker of physical functionality (i.e., risk of fall). 

With respect to the study protocol and Table 1, we found no article that analyzed vitamin D exposure in populations in association with other dimensions of healthy ageing as well-being and quality of life, environmental and financial resources, personality, health status, and ageing self-perception.

### 2.6. Risk of Bias in Individual Studies

Quality evaluation was made using the Newcastle Ottawa scale (NOS) for cohort studies and case-control (for nested case-control studies). The evaluation was conducted by two reviewers (SC, TT) in blind, and discordances were discussed.

### 2.7. Data Synthesis

A synthesis of data was performed after qualitative analysis carried out according to the methods of the study (SC, AS). Characteristics of cohorts and participants were reported in a table of the characteristics of the studies (Appendix A Characteristics of studies included); vitamin D congener, outcomes and qualitative synthesis of results were reported in the table of results (Appendix A table of results).

## 3. Results

A total of 24 studies were selected for inclusion in this review (S2 Studies included). Cumulatively, the search on PubMed, Scopus, and Web of Science databases returned a total of 3960 records. After deduplication, 3172 records remained. Of these, 2836 were excluded by title and abstract for their inconsistency with the inclusion criteria. Subsequently, 336 full-texts were read, and the following were excluded: 80 articles for characteristics of study design; 28 articles for incoherence with age criteria; 139 because on unhealthy people at baseline; 16 due to vitamin D supplement intake at baseline. In 4 other articles vitamin D was not the exposure factor analyzed, outcomes used for the purpose of this SR were lacking in 5 articles, and 36 articles failed to meet more than one of inclusion criteria. Finally, 4 more citations were excluded because they did not report analysis results (Figure 1).

The 24 articles included in this SR contain data from 20 cohorts, encompassing a total of 77,629 individuals (48% women). List of cohorts and articles is available in the annex (Appendix A Studies included), as well as the table reporting the characteristics of observed populations (Appendix A Characteristics of studies included). Of the 24 articles, 5 presented nested case-cohort analyses [55,56,57,58,59]. Follow-up of the studies varied from 1 to 26 years since baseline [60,61].

Half of the studies were from European countries (Denmark, Finland, Netherland, Sweden, UK, Spain, and Switzerland), and a quarter of them was from the US. Other cohorts were from Australia (*n* = 2), Japan (*n* = 2), and Lebanon (*n* = 1). Mean age at baseline ranged from 51 [56] to 79 years old [61]. All studies recruited apparently healthy people (with a proportion of participants with chronic diseases <25% of the entire study sample) who did not use vitamin D supplements at baseline.

Studies presented analyses on longitudinal associations between vitamin D levels at baseline and the outcomes herein selected. While 20 studies presented analyses exclusively on 25(OH)D, 4 also reported on 1,25(OH)_2_D levels [55,62,63,64], and one only on this latter biomarker [65]. The predictivity was estimated as hazard ratios (HRs) or odds ratios (ORs) by dividing participant levels of vitamin D at baseline in percentile categories and using the lowest or the highest percentile as reference, whereas four studies calculated HRs/ORs for SD increases in serum 25(OH)D or 1,25(OH)_2_D [58,59,63,66]. 

The primary outcomes of the included studies were mainly related to disease incidence (87%), all-cause mortality (9%), cognitive and physical functionality (1.8% for both). We did not find any analyses on the other six dimensions of healthy ageing concept (Table 1). Data were collected on 65 associations between vitamin D and healthy ageing outcomes, with 11 associations relating to 1,25(OH)_2_D. Diseases outcomes were the incidence of cardiovascular (CV), coronary, cardiometabolic, cancer, pulmonary, and respiratory events; osteoporosis; sarcopenia; dementia; Alzheimer’s disease. Physical functionality outcome was reported only by one study (risk of fall), as well as only one study reported on cognitive dimensions (25). Associations between 25(OH)D and osteoporosis (21 associations) or CV/coronary events (14 associations) were the most frequently observed. 

### 3.1. Risk of Bias within Studies

The largest number of studies achieved a NOS score ≥8: 8 studies achieved the top score of 9, while 10 studies a score of 8. Among the remaining five studies, scores ranged from 5 to 7 (Table 2). Collectively, the quality of studies included in this SR was medium-high. A total of 10 studies [55,56,61,63,64,66,67,68,69,70] did not include data on the adequacy of follow-up in relation to the proportion of people completing the studies. Seven studies failed to mention that the outcome of interest was absent at enrolment [59,61,62,63,64,71,72]: two were nested case-control studies [59,72]. Finally, one study [73] failed to express categories cutoffs for vitamin D used for analyses, and another [61] did not explicitly mention the independent process of outcome evaluation.

### 3.2. Predictivity of 25(OH)D Levels

All results are presented in the annexed Appendix A table of results. Table 3 presents numbers and direction of associations between 25(OH)D concentration at baseline and the outcomes selected for this SR. We found three positive associations, 31 negative associations, and 19 non-significant differences between concentrations of 25(OH)D at baseline and the risk of an event at follow-up. 

#### 3.2.1. All-Cause Mortality 

Five studies presented inverse associations of serum 25(OH)D concentrations and risk of death for all causes [57,60,68,69,74]. The risk of premature death was >30% [75,76] for the lowest categories of vitamin D levels compared to the highest one. Brøndum-Jacobsen et al. [74] used the Kaplan-Meier curve of survival as indicator of early death confirming negative association with mortality. Lastly, Khaw et al. [68] and Heath et al. [57] showed similar associations: an 11% and 14% reduction in the risk of death for a 20 and 25 nmol/L increase in serum 25(OH)D concentrations, respectively.

#### 3.2.2. Cardiovascular, Coronary, and Cardiometabolic Events

These events were observed in five articles reporting on negative associations between 25(OH)D and CV or coronary events. The risk of CV events increased with declining vitamin D levels at baseline. Liu et al. [69] reported a 2-time higher risk of death from heart failure for the lowest tertile in comparison with the highest one, while Brøndum-Jacobsen et al. [74] showed a >50% higher risk of the lowest 25(OH)D category, which was up to 110% for fatal events. In three studies, evidence of a protective effect for the highest vitamin D levels was found [56,59,68], with a nearly 50% risk reduction between the highest and lowest quartile [56] and with a HR = 0.71 for every increase of a SD in vitamin D level [59]. Finally, 25(OH)D was not associated with CV mortality [63] nor with non-fatal CV events [74]. Al-Khalidi et al. showed an inverse association with cardiometabolic causes of mortality (i.e., heart diseases, cerebrovascular diseases, and diabetes mellitus), concluding that total levels of serum vitamin D <30 nmol/L were predictive of high lifetime risk of cardiovascular and metabolic death, even without weight loss or BMI modifications [77]. By contrast, 25(OH)D was not associated with CV mortality [63], non-fatal CV events [74], or insulin resistance [70].

#### 3.2.3. Impaired Bone Health

Six articles report data on 17 associations between vitamin D and bone health outcomes [58,64,66,67,68,71]. In three articles [64,67,71], bone density and bone loss were assessed at different sites (lumbar spine, hip and femoral neck, trochanter, forearm). All studies found no association between 25(OH)D and biomarkers of bone. Arabi et al. [67] reported a significant predictivity for high parathyroid hormone (PTH), but not for low vitamin D. However, Swanson et al. presented evidence of a negative association for bone loss at the hip and lumbar spine, underlining a lower bone loss for each SD increase of vitamin D concentration at baseline [64]. A negative association was shown by these authors for vitamin D with hip fracture incidence. In three other studies, outcomes relating to fractures at the hip and non-vertebral sites were reported [64,66,68]. The first one found an 11% higher risk of people with vitamin D <30 nmol/L in multivariate-adjusted models, with risk totaling 19% when hip fractures were considered [68]. A negative association with all type of fractures was also shown in this study. The others reported, for each SD increase in 25(OH)D, a 30% reduction of the hip fracture risk and a similar decrease (26 %) of the incidence of major fractures [64,66]. Associations with non-vertebral fractures were not statistically significant [64].

#### 3.2.4. Respiratory Events

Three articles reported results of 25(OH)D associations with respiratory outcomes. Afzal et al. [73] showed positive associations with forced expiratory volume in the 1st second (FEV1) and forced vital capacity (FVC), with results being not statistically significant among non-smokers. In the same study, a negative association was observed between 25(OH)D and chronic obstructive pulmonary disease (COPD). While Aregbesola et al. [78] found a negative association with the risk of hospitalization for pneumonia, Khaw et al. observed similar associations with respiratory events and respiratory death [68].

#### 3.2.5. Cancer

Three studies showed positive associations of 25(OH)D with the risk of cancer or cancer mortality [55,68,79]. All associations were statistically not significant, except for one study by Afzal et al. [79] reporting a positive association with non-melanoma skin cancer and another by Khaw et al. [68] who observed an 11% reduction of cancer mortality for each 20 nmol/L increase in 25(OH)D.

#### 3.2.6. Sarcopenia

Hirani et al. showed a significantly increased risk of hospitalization for sarcopenia, with 140% higher incidence for the lowest quartile 25(OH)D compared to the highest one (<40 vs. ≥69 nmol/L) [62].

#### 3.2.7. Neurological Diseases and Cognitive Functionality

Results for the risk of dementia were discordant. While Olsson and colleagues did not show any association [80], Licher et al. found a 10% higher risk of dementia for people in the lowest 25(OH)D category in comparison to the highest, as well as a 13% higher risk of developing Alzheimer’s [81]. Only one study reported on associations with markers of cognitive impairment, showing no difference across vitamin D levels [80]. 

#### 3.2.8. Secondary Outcome

The unique article which deals with fall incidence showed an inverse association with 25(OH)D concentrations at baseline [61].

### 3.3. Predictivity of 1,25(OH)_2_D Levels

The number of studies focusing on 1,25(OH)_2_D predictivity was remarkably lower than for 25(OH)D [(five articles for 1,25(OH)_2_D vs. 24 for 25(OH)D)] [55,62,63,64,65]. Table 4 summarizes the numbers and direction of associations found for this metabolite at baseline when plotted against the different outcome measures. We found 11 associations for 1,25(OH)_2_D levels. Six were negative and pinpointed a significantly increased risk of events along with declining 1,25(OH)_2_D levels at baseline. The remaining five associations were not significant. The consistency of results for cardiovascular/coronary events and outcomes was low.

The study by Umehara et al. was the only one presenting a negative relationship between 1,25(OH)_2_D and all-cause mortality: the first quartile showed a 54% higher risk compared to the highest one [65]. The same authors described an inverse association with CV mortality [65], which was similarly documented in a study by Jassal et al. [63]. Also significant was the association with respiratory death, but not with cancer mortality [65].

The association between skeletal outcomes and 1,25(OH)_2_D was weaker than that with 25(OH)D [64]. In particular, Swanson et al. found no association between 1,25(OH)_2_D and bone loss at the hip and lumbar spine [64] and a negative relationship with hip fractures. With regards to sarcopenia, the lowest 1,25(OH)_2_D concentrations were associated with a 120% higher risk of hospitalization for sarcopenia at 2- and 5-years follow-up in men aged 70 years and over [62]. Lastly, no significant association was found by Platz et al. for cancer events [55].

## 4. Discussion

This systematic review was conducted to assess and ascertain the ability of vitamin D metabolites [25(OH)D and 1,25(OH)_2_D] to act as promising predictive biological biomarkers of the multidimensional process of ageing. A considerable number of previous systematic reviews and meta-analyses has analyzed and discussed the association between circulating levels of vitamin D and only one of the dimensions involved in the genesis of the ageing process and described in detail by Depp et al. [32]. Opposed to most previous studies, which included all age ranges as well as any health status (e.g., healthy and unhealthy), our study was focused on apparently healthy cohorts aged >50 years.

We found 24 articles reporting data from 20 cohort studies with a total of 77,629 participants, and 54 associations between 25(OH)D concentration at baseline and health outcomes, as well as 11 associations for 1,25(OH)_2_D, were analyzed. The number of studies on 25(OH)D predictivity appeared remarkably greater than for 1,25(OH)_2_D and can be due to the general consensus existing on 25(OH)D measurement as a reliable marker of vitamin D status and, possibly, the higher costs of routinely measuring 1,25(OH)_2_D. 

The most relevant result is represented by the strong association between a low level of 25(OH)D and higher all-cause mortality considered to be a strong surrogate of longevity, with large and significant effect sizes in all five cohorts included. 

On the other hand, the most explored associations for 25(OH)D were with cardiovascular and coronary events (12 associations reported by five articles) and with bone loss or fractures (12 associations reported by six articles); in both cases an inverse effect was shown, even if, for bone health indicators, the effect sizes and the consistency between studies were lower. While association of low levels of vitamin D with all-cause mortality [82,83] and cardiometabolic risk [84,85,86] were confirmed by previous reviews, our results do not support a role for vitamin D as a predictor of cancer risk, particularly prostate and colorectal cancer [87,88,89].

Like our, a previous study [3] reviewed the predictive role of vitamin D on several outcomes representative of healthy ageing. However, such study focused on cohorts that were unhealthy at baseline, not allowing extension of the study conclusions to the general population.

Our review included numerous markers of the state of ageing, to circumvent the lack of a single reliable marker of healthy ageing in the literature [32], but we partially failed in this objective. Other systematic reviews have addressed the relationship between circulating vitamin D and frailty [15,90], depression incidence [91], cognitive functionality [92], and physical performance [93], but we were not able to find adequate eligible studies on these outcomes in keeping with our selection criteria.

Among the studies herein reviewed, two reported data for multiple outcomes relating to disease incidence and overall mortality [65,68], whereas data on several healthy ageing dimensions were collected from the Copenhagen City Heart Study, where pulmonary function, cancer, early death, and cardiovascular events were analyzed [73,74,94], and from the Third National Health and Nutrition Examination Survey (NHANES III), exploring results on premature death, CVD mortality, and cardiometabolic mortality [69,77].

Although these analyses investigated healthy cohorts for more than one outcome, the limited incidence of early and/or premature death as well as of chronic CVD or cancer hampered our ability to draw definitive conclusions on vitamin D status predictivity in these specific dimensions. Likewise, studies evaluating outcomes related to physical functionality failed to assess ADL/IADL scales or disability incidence, thus not allowing assessment of vitamin D effects on core physical functionality tests. Furthermore, to the best of our knowledge, there were no studies published on social functionality or other dimensions of healthy ageing such as personality traits, well-being, self-perception of health or ageing status, and quality of life [32].

This work has some weaknesses that must be taken into account. First, the studies analyzed predominantly dealt with Caucasian populations, thus not allowing extension of current results to non-Caucasian populations. Secondly, studies were excluded if not collecting health status at baseline, and the inclusion of these possibly relevant studies could have led to potentially different conclusions. The heterogeneity of indicators used by the studies prevented from carrying out meta-analyses of results. 

Nevertheless, the estimates of the present work are based on longitudinal studies, thus avoiding cross-sectional bias and increasing the scientific confidence in causality. Also, differently from the other reviews on the topic, this study explored the relationship between vitamin D metabolites and all the multiple components of the ageing of the general population on the basis of studies judged as high quality, with a NOS score ≥8. 

## 5. Conclusions

Conclusively, 25(OH)D status acts as a worthy biomarker to predict all-cause mortality, pulmonary events, and lung function, cardiovascular/coronary events, hip fractures, early and/or premature death, and sarcopenia incidence. Conversely, no association was found with the risk of falls, as well as with bone mineral density, cognitive function, cardiometabolic, and cancer events in adult populations who were apparently healthy at baseline. As far as 1,25(OH)_2_D is concerned, its associations with all-cause mortality and respiratory events do not seem to be of relevant clinical value. 

There is a clear limited scientific evidence on the role of vitamin D metabolites in the multidimensional process of ageing, since the studies analyzed by our review deal with a limited spectrum of markers of healthy ageing, and markers such as social participation, social function, as well as subjective dimensions related to self-perception of own status and resources [95,96,97,98] were missing. For this reason, we cannot draw definitive conclusions on vitamin D as a reliable predictor of the healthy ageing process for all dimensions herein investigated. Nevertheless, we are inclined to consider the robust results on relevant markers such as longevity and incidence of disease, as a strong support to consider vitamin D as a multidimensional predictor of ill health in the ageing process. Further well-designed controlled trials to investigate whether vitamin D supplement results in superior outcomes are warranted in the future.

## Figures and Tables

**Figure 1 nutrients-11-02384-f001:**
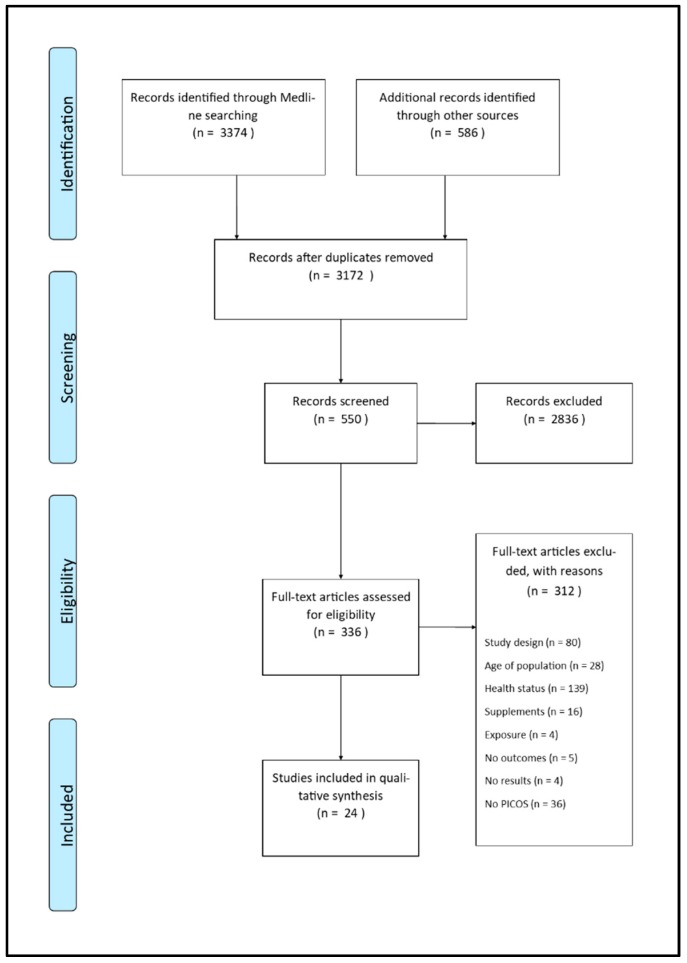
PRISMA flow diagram. The flow chart below represents the selection process of studies included in this systematic review, from the identification of records to the inclusion phase.

**Table 1 nutrients-11-02384-t001:** Dimensions and definitions of healthy ageing concept and instruments for measurements. Following a systematic review [32], table reports on the first ten dimensions of healthy ageing concept more frequently analyzed in clinical studies on ageing.

Dimension of Healthy Ageing	Definition	Scale, Questionnaire, Instruments
Longevity	Physiological ability to survive more than the mean expectancy life	All-causes death Mortality for specific causes Survival
Lack of diseases	Lack of diagnosed pathologies	Incidence of diseases such as cardiovascular and cerebrovascular events, diabetes, dementia, and Alzheimer’s, respiratory diseases, etc. or the Healthy Life Years
Physical functionality	Ability and autonomy in daily functional tasks [33]	Activities of Daily Living (ADL)/Instrumental Activities of Daily Living (IADL) scales [32], [34,35,36], falls and fear of fall, or physical performance test (i.e., Time Up & Go Test,) [32,37], SF-36 Physical Function/Role Function Scales [38,39,40]
Cognitive functionality	Complex ability related to speech, understanding, memory, learning, attention and concentration, reading and writing abilities, calculation ability, opinion, planning ability, problem-solving, etc.	Cognitive screening test (MMSE), SPMSQ score [41], cognitive abilities measures [42], (Mini) Mental Status Test [43], executive function [44]
Social functionality	Ability to preserve rules and responsibilities in the different social (formal and informal, productive and unproductive) environments	Modality to pass leisure time, monthly contacts with friends and familiars, participation in social aspects of community, visit friends and familiars, social support, payed work and care of children/partner [32]
Well-being and quality of life	Mental status related to all positive and negative evaluation and emotive reactions to lived experiences	Cantril’s Ladder of Life Scale [45], Scale of Positive and Negative Experience (SPANE) [46], the European Social Survey well-being [47], the Life Satisfaction scale/Life Satisfaction Index (LSI) [48,49], the CES-D [50], Psychological Well-Being Scale (PWB) [46], the Flourishing index [51] or the Geriatric Depression Scale [52]
Perceived health status	Perception of own health status	General Health Questionnaire 0–5 [53] or SF-36 Quality of Life questionnaire
Personality	Structured modality of thought, feeling, and behavior resulting from interaction between environments, genetic makeup, and cultural heritage	Test of perceived control [32] or with the Extraversion (6-item) and goal strength [54]
Resources and environment	Security sense comes from financial and social environment	Salary and financial security [32]
Ageing status perceived	Perception of own ageing status or sense given to ageing	Likert scales [32]

**Table 2 nutrients-11-02384-t002:** Total Newcastle Ottawa Score (NOS) for studies included in the SR. NOS ranges from 0 (low quality) to 9 (best quality). NOS scale aims to assess the quality of studies in three domains related to the selection process: comparability of cohorts and outcome assessment and adequacy. Total score resulted by two-blinded reviewers’ agreement.

Author	Cohort/Study	Total Score (NOS)
Afzal 2014	Copenhagen City Heart Study & Copenhagen General Population Study	8
Afzal 2013	Copenhagen City Heart Study	9
Al-khalidi 2019	Third National Health and Nutrition Examination Survey (NHANES III)	9
Arabi 2012	Lebanese cohort	7
Aregbesola 2013	Kuopio Ischaemic Heart Disease Risk Factor Study (KIHD)	8
Barrett-Connor 2012	Osteoporotic Fractures in Men Study (MrOS)	8
Brøndum-Jacobsen 2012	Copenhagen City Heart Study	9
Cauley 2010	Osteoporotic Fractures in Men Study (MrOS)	8
Heath 2017	Melbourne Collaborative Cohort Study (MCCS)	9
Hirani 2018	Concord Health and Ageing in Men Project (CHAMP)	7
Holmberg 2017	Swedish farmers study	8
Jassal 2010	The Rancho Bernardo Study	6
Khaw 2014	EPIC cohort	8
Licher 2017	The Rotterdam Study	9
Liu 2012	Third National Health and Nutrition Examination Survey (NHANES III)	8
Looker 2013	NHANES III & NHANES 2000–2004	8
Marques-Vidal 2015	Cohorte Lausannoise (CoLaus) study	7
Mursu 2015	Kuopio Ischaemic Heart Disease Risk Factor Study (KIHD)	9
Olsson 2017	Uppsala Longitudinal Study of Adult men	9
Platz 2004	Health Professionals Follow-up Study	8
Shimizu 2015	Otasha–Kenshin study	5
Swanson 2015	Osteoporotic Fractures in Men Study (MrOS)	7
Umehara 2017	Hisayama study	9
Vàzquez-Oliva 2018	REGICOR (Registre Gironı del COR) population cohort study	8

**Table 3 nutrients-11-02384-t003:** Numbers and direction of associations between 25(OH)D and the outcomes selected for this SR. Positive associations (or direct associations) are those when risk of event measured at follow-up increased with the increase of 25(OH)D levels at baseline; Negative (or inverse associations) are those representing an increased risk of event incidence at follow-up with the decrease of 25(OH)D levels at baseline; the third column reports non-statistically significant associations between 25(OH)D levels at baseline and risk of an event at follow-up.

Outcomes	Positive Association	Negative Association	No Statistically Significant Association	Total
All-cause mortality	0	5	0	5
Pulmonary and respiratory events	2	5	0	7
Cancer events	2	0	3	5
Cardiovascular and coronary events	0	9	3	12
Cardiometabolic events	0	1	1	2
Impaired bone health	0	7	10	17
Sarcopenia	0	1	0	1
Dementia and Alzheimer’s	0	2	1	3
Physical functionality (falls)	0	1	0	1
Cognitive functionality	0	0	1	1
Total	3	31	19	54

**Table 4 nutrients-11-02384-t004:** Number and direction of associations between 1.25(OH)D and outcomes selected. Negative (or inverse associations) are those representing an increased risk of event incidence at follow-up with the decrease of 25(OH)D levels at baseline; the second column reports non-statistically significant associations between 25(OH)D levels at baseline and risk of an event at follow-up.

Outcomes	Negative Association	No Statistically Significant Association	Total
All-cause mortality	1	0	1
Pulmonary and respiratory events	1	0	1
Cancer events	0	2	2
Cardiovascular and coronary events	2	0	2
Bone health	1	3	4
Sarcopenia	1	0	1
Total	6	5	11

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
