# Peer review of "Vitamin D as a Biomarker of Ill Health among the Over-50s: A Systematic Review of Cohort Studies"

_nutrients, 2019, doi:10.3390/nu11102384_

Round 1

Reviewer 1 Report

Dear editors:  

 Silvia Caristia and co-authors conducted a systematic review of the Vitamin D as “a biomarker of healthy ageing” among the over-fifties. I have a number of questions and comments concerning this study:

World Health Organization (WHO) defines “health” as including not only physical and mental health, but also social well-being in various domains. And“healthy ageing” is interchangeably with terms such as “active”, “successful”, or “productive ageing”. “A biomarker of healthy ageing” is reminiscent of surrogate markers or scales that have been used to assess healthy aging in research. However, primary measures of 20 studies included in this systemic review are not relevant to the assessment of healthy aging (Text and Tables). The term“healthy ageing” never exists in purposes, outcomes, and conclusions of the selected studies, although authors intend to conduct a systemic review about predictive values of vitamin D for healthy ageing using a string and MeSH Terms of aged, healthy ageing, vitamin D and its synonymous. As mentioned above, based on the combined results of the review, it is not reasonable to conclude that “Our study provides an update on the predictive role of vitamin D status on future healthy ageing. Current results can contribute to increase current knowledge on vitamin D actions in physiologic/pathologic ageing processes to improve frailty prevention in ageing.” The review did not address a clearly focused question of healthy ageing. The authors did not look for the right type of papers about ageing issues. According to current scientific evidence, there has been no identification of a single biomarker or gold standard tool that can monitor healthy aging successfully [Wagner KH, et al. Biomarkers of Aging: From Function to Molecular Biology. Nutrients. 2016 Jun 2;8(6). pii: E338.][ Lara J, et al. A proposed panel of biomarkers of healthy ageing. BMC Med. 2015;13:222.][ Bürkle A, et al. MARK-AGE biomarkers of ageing. Mech Ageing Dev. 2015 Nov;151:2-12.][Wentian Lu, et al. Domains and Measurements of Healthy Aging in Epidemiological Studies: A Review, The Gerontologist, Volume 59, Issue 4, August 2019][…]. Thus a biomarker of healthy ageing based solely on circulating concentrations of Vitamin D is insufficient to assess“healthy ageing”. Age linearly correlates with and/or is caused by the accumulation of reactive oxygen species, DNA damage, mitochondrial dysfunction, impaired antioxidant defense and shortening of the telomeres. Emerging evidence of molecular/ DNA-based novel biomarkers for aging are well established in humans. Corresponding to the previous question, a more comprehensive systemic review should include Vitamin D studies with novel parameters for aging [Schöttker B, et al. Serum 25-Hydroxyvitamin D Levels as an Aging Marker: Strong Associations With Age and All-Cause Mortality Independent From Telomere Length, Epigenetic Age Acceleration, and 8-Isoprostane Levels. J Gerontol A Biol Sci Med Sci. 2019 Jan 1;74(1):121-128.]. Except for the issue of healthy ageing assessment, I think the authors do enough to assess the quality of the included studies. With respect to the previous systematic review about vitamin D [Autier P, Boniol M, Pizot C, Mullie P. Vitamin D status and ill health: a systematic review. Lancet Diabetes Endocrinol. 2014 Jan;2(1):76-89.], low circulating levels of vitamin D is actually a warning sign of ill health. In the present study, more recent studies were reviewed with updated viewpoints. In my opinion, the authors should rephrase their study and resubmit after major revision. For example, the title could be rephrased as “Vitamin D as a versatile biomarker of ill health among the over-fifties: a systematic review of cohort studies.” Since all the outcome studies are not relevant to the assessment of healthy aging, the words concerning healthy ageing should be avoided in the article. In the section of 3.2.4. Bone and muscle health, authors address that the risk of major fractures increased approximately by 26% for each SD decrease in 25(OH)D. Considering the dominant theory of vitamin D deficiency related bone events, the sentence in the abstract “whereas no association was documented for osteoporosis and cognitive outcomes” should be deleted. Shall we clarify the association between vitamin D and fracture risks/ sarcopenia in the abstract?

Author Response

Answers to Reviewer #1 comments:

World Health Organization (WHO) defines “health” as including not only physical and mental health, but also social well-being in various domains. And“healthy ageing” is interchangeably with terms such as “active”, “successful”, or “productive ageing”. “A biomarker of healthy ageing” is reminiscent of surrogate markers or scales that have been used to assess healthy aging in research. However, primary measures of 20 studies included in this systemic review are not relevant to the assessment of healthy aging (Text and Tables). The term“healthy ageing” never exists in purposes, outcomes, and conclusions of the selected studies, although authors intend to conduct a systemic review about predictive values of vitamin D for healthy ageing using a string and MeSH Terms of aged, healthy ageing, vitamin D and its synonymous.

We thank the Reviewer for his comments on current concepts of healthy ageing and biomarkers capable of reflecting its trajectories. We share the Reviewer’s view on the interchangeable terminology relating to healthy ageing. We decided to choose this expression to better reflect the “ageing process while developing without major loss in health and functionality”. While the expression “healthy aging” does not exist in many texts of the studies here selected, the same applies to any other term referred to ageing and mentioned in the review, such as active, successful or productive. However, the introduction of the revised manuscript has been changed to integrate such criticisms in a constructive manner (lines 34-43).

As mentioned above, based on the combined results of the review, it is not reasonable to conclude that “Our study provides an update on the predictive role of vitamin D status on future healthy ageing. Current results can contribute to increase current knowledge on vitamin D actions in physiologic/pathologic ageing processes to improve frailty prevention in ageing.” The review did not address a clearly focused question of healthy ageing.

With concern to our focusing on vitamin D as a biomarker of healthy ageing, our manuscript focused on vitamin D is intended to be published on a special issue of Nutrients on vitamin D. This secosteroid is particularly relevant because it acts both as an endogenous marker and as a potential exogenous/therapeutic agent - a condition that partly applies to other hormones acting with an age-related pattern. We are certainly aware that any research focused on a single marker of ageing is self-limiting, and decided to change the paper conclusions to accommodate these judicious criticisms (lines 27-30 and 404-412).

The authors did not look for the right type of papers about ageing issues. According to current scientific evidence, there has been no identification of a single biomarker or gold standard tool that can monitor healthy aging successfully [Wagner KH, et al. Biomarkers of Aging: From Function to Molecular Biology. Nutrients. 2016 Jun 2;8(6). pii: E338.][ Lara J, et al. A proposed panel of biomarkers of healthy ageing. BMC Med. 2015;13:222.][ Bürkle A, et al. MARK-AGE biomarkers of ageing. Mech Ageing Dev. 2015 Nov;151:2-12.][Wentian Lu, et al. Domains and Measurements of Healthy Aging in Epidemiological Studies: A Review, The Gerontologist, Volume 59, Issue 4, August 2019][…]. Thus a biomarker of healthy ageing based solely on circulating concentrations of Vitamin D is insufficient to assess“healthy ageing”. Age linearly correlates with and/or is caused by the accumulation of reactive oxygen species, DNA damage, mitochondrial dysfunction, impaired antioxidant defense and shortening of the telomeres. Emerging evidence of molecular/ DNA-based novel biomarkers for aging are well established in humans. Corresponding to the previous question, a more comprehensive systemic review should include Vitamin D studies with novel parameters for aging [Schöttker B, et al. Serum 25-Hydroxyvitamin D Levels as an Aging Marker: Strong Associations With Age and All-Cause Mortality Independent From Telomere Length, Epigenetic Age Acceleration, and 8-Isoprostane Levels. J Gerontol A Biol Sci Med Sci. 2019 Jan 1;74(1):121-128.]. Except for the issue of healthy ageing assessment, I think the authors do enough to assess the quality of the included studies. With respect to the previous systematic review about vitamin D [Autier P, Boniol M, Pizot C, Mullie P. Vitamin D status and ill health: a systematic review. Lancet Diabetes Endocrinol. 2014 Jan;2(1):76-89.], low circulating levels of vitamin D is actually a warning sign of ill health. In the present study, more recent studies were reviewed with updated viewpoints. In my opinion, the authors should rephrase their study and resubmit after major revision. For example, the title could be rephrased as “Vitamin D as a versatile biomarker of ill health among the over-fifties: a systematic review of cohort studies.” Since all the outcome studies are not relevant to the assessment of healthy aging, the words concerning healthy ageing should be avoided in the article.

The selection process that was followed according to the strict inclusion criteria (healthy population at baseline, young adult age, lack of treatment) led us to exclude an enormous number of publications. This may explain why several studies could not be included in this systematic review. With concerns to the study published by Schöttker et al., this was excluded because in baseline conditions approximately 62% of the female cohort was undergoing estro-progestinic therapy and the rate of hypertensive patients within the entire cohort outnumbered that quoted in our inclusion criteria. With regards to Autier’s paper, this was included both in the original submission (ref. 10) and in this revised form (ref. 7), and hence discussed (lines 364-367). Considering the comments and in agreement with the Reviewer’s suggestion, we have now rephrased the title in the revised version of the manuscript.

In the section of 3.2.4. Bone and muscle health, authors address that the risk of major fractures increased approximately by 26% for each SD decrease in 25(OH)D. Considering the dominant theory of vitamin D deficiency related bone events, the sentence in the abstract “whereas no association was documented for osteoporosis and cognitive outcomes” should be deleted. Shall we clarify the association between vitamin D and fracture risks/ sarcopenia in the abstract?

We thank the Reviewer for this comment, and decided to modify the abstract accordingly (lines 25-56)

Reviewer 2 Report

 Vitamin D as a biomarker of healthy ageing among the over-fifties: a systematic review of cohort studies

The topic of this paper is interesting. However, there are several shortcomings and inconsistencies which need to be revised. There are also many parts with complicated sentences or bad language. The results of the study do not allow the definite conclusions the authors have drawn.

Abstract

Line 15 Why do you mention explicitely extra-skeletal chronic diseases? This leads to the impression that you will focus on extra-skeletal diseases only which is obviously not the case.

Lines 15-17 Sentence should be revised – low vitamin D as marker for healthy ageing promotion?

Line 18 …cohorts observing the role of

Line 22 …were evaluated… to be deleted

Line 23 Twenty cohorts but 24 studies were included

Introduction

Line 44  …in mammalian ageing…

Line 44 “Elderly” should be replace by older adults

Line 47 institutionalized older adults

Line 54 “multiple sclerosis” is not a typical disease of older people, but typically presents with 30/40 years

Line 55 A higher vitamin D status is associated…

Lines 61-65 Very long sentence mixing results in C. elegans with expected cohort results. Should be revised.

Lines 65-67 should be expressed more carefully, e.g. might prevent… (References?)

Line 76 …of cohort and longitudinal studies… Usually cohort studies are longitudinal studies.

Materials and Methods

Why was the SR not registred in PROSPERO?

Line 92 …except for vitamin D – Does this mean participants were allowed to take vitamin D supplements but no other ditary supplements?

Line 93 Sentence: less than 25% of subjects with prevalent diagnosis?

Line 94 … or taking vitamin D supplements – here only 25% of persons were allowed to take vitamin D in a sample – contradictory to line 92?

Line 139 independently instead of blindly?

Line 141 …which passed the first step

Line 143 …full-text selection was registered with reasons for exclusion – not clearly expressed

Line 159 Please explain the choice why these are secondary outcomes as physical performance is also a primary outcome (ADL scale) – hardly possible to differentiate

Line 161 …we found…

Results

Table 3: What about bone health? Here you report both in one line (physical functionality with falls).

Line 248: As you report ranges before, it would be interesting to know the ranges of the two highest quintiles and the lowest quintiles.

Line 249 This seems to be missing in Table 3?

Line 255 ….>50% higher risk…

Line 259 not associated with CV…

Line 263 …heart diseases, cerebrovascular diseases can hardly be separated from 3.2.2. How did you manage this?

Section 3.2.4 on bone and muscle health does not include any information on muscle health

Line 299 …with marker of …?

Section 3.2.9 may be shifted to 3.2.4, what about the other secondary outcomes? Were there no results?

Line 310 What is meant by physical functionality? It seems to be missing in Table 4

Discussion

Large part of the discussion repeats the results.

Line 326 ..between vitamin D and biomarkers?? Mostly health outcomes/diseases?

Line 327 see comment on Line 76

334 …as low quality

Line 356 ...negative associations were observed with all-cause…?

Line 364 …. in young subjects – what does this mean in the context of premature death?

Line 371 However, this study focused on cohort with a high proportion of unhealthy participants?

Line 373 Based on the previous evidence,…

Lines 376/377 what do you mean by: this is clearly related to the different types of cancer highlighted in studies selected for the current review.

Line 392 …on cohorts with healthy younger adults…

Conclusions

These definite conclusions cannot be drawn from the results of this systematic review. For example, there were only two studies on sarcopenia included and only one with 25(OH)D. Also on pulmonary events, there were different outcomes and one association was only significant in smokers. This section needs to be revised and more cautious conclusions should be drawn from the results.

In general

vitamin D instead of vitamin-D

1,25(OH)D or 1.25(OH)D

Author Response

Answers to Reviewer #2's comments

Abstract

Line 15 Why do you mention explicitly extra-skeletal chronic diseases? This leads to the impression that you will focus on extra-skeletal diseases only which is obviously not the case.

We agree with the reviewer. The vitamin D deficiency is strictly related to a considerable number of chronic diseases, not only extra - skeletal. We corrected the sentence (line 15). 

Lines 15-17 Sentence should be revised – low vitamin D as marker for healthy ageing promotion?

We agree with the Reviewer, this sentence has been corrected (line 17).

Line 18 …cohorts observing the role of

Thanks for the suggestion, this has been corrected (line 19)

Line 22 …were evaluated… to be deleted

Thanks for the suggestion, this has been corrected (line 22)

Line 23 Twenty cohorts but 24 studies were included

Thanks for the suggestion, this has been corrected (line 23)

Introduction

Line 44  …in mammalian ageing…

Thanks for the suggestion, this has been corrected (line 55)

 Line 44 “Elderly” should be replace by older adults

This has been done (line 55)

Line 47 institutionalized older adults

Thanks for the suggestion, this has been corrected (line 58-59)

Line 54 “multiple sclerosis” is not a typical disease of older people, but typically presents with 30/40 years

We agree with the Reviewer. This statement is not relevant for the study purpose and has been deleted.

Line 55 A higher vitamin D status is associated…

Thanks for the suggestion, this has been deleted (lines 65-66)

Lines 61-65 Very long sentence mixing results in C. elegans with expected cohort results. Should be revised.

We agree with this observation and thank the Reviewer for pointing out this verbose sentence. We have modified the sentence accordingly (line 69-72).

Lines 65-67 should be expressed more carefully, e.g. might prevent… (References?)

We changed modal verb of possibility (from should to might) and we simplified the sentences (line 71-72)

Line 76 …of cohort and longitudinal studies… Usually cohort studies are longitudinal studies.

We agree with the reviewer. We aimed to mention panel studies since longitudinal studies may not all reflect cohorts. This has been corrected (line 80)

Materials and methods

Why was the SR not registered in PROSPERO?

Unfortunately we didn't have the time due to submission timing rules for the special issue on vitamin D , even if the protocol is available upon request.

Line 92 …except for vitamin D – Does this mean participants were allowed to take vitamin D supplements but no other dietary supplements?

This was a mistake, the sentence has been now corrected (lines 92)

Line 93 Sentence: less than 25% of subjects with prevalent diagnosis?

We thanks the Reviewer for the suggestion, this has been corrected (lines 93-94)

Line 94 … or taking vitamin D supplements – here only 25% of persons were allowed to take vitamin D in a sample – contradictory to line 92?

This sentence is actually correct, line 92 contained a mistake that has now been corrected (lines 92-96)

Line 139 independently instead of blindly?

Thanks for the suggestion, this has been corrected (line 142)

Line 141 …which passed the first step

Thanks for the suggestion, this has been corrected (line 144)

Line 143 …full-text selection was registered with reasons for exclusion – not clearly expressed

Thanks for the suggestion, we rephrased the sentence (line 147).

Line 159 Please explain the choice why these are secondary outcomes as physical performance is also a primary outcome (ADL scale) – hardly possible to differentiate

Thanks for the comment, a sentence has been added to clarify this point (lines 162-167)

Line 161 …we found…

Thanks for the suggestion, this has been corrected (line 168)

Table 3: What about bone health? Here you report both in one line (physical functionality with falls).

There was a mistake for lacking of coherence between text and table 3. Outcomes under the “Bone Health” title are biomarkers of osteoporosis. We changed the title for these types of outcome from osteoporosis to Bone health in Table 3 and text (lines 275-276). We also deleted “muscle” from the title as pointed out by the Reviewer (line 274)

Line 248: As you report ranges before, it would be interesting to know the ranges of the two highest quintiles and the lowest quintiles.

Thanks for the suggestion, the sentence has been actually changed (line 251-255).

Line 249 This seems to be missing in Table 3?

There was a mistake for lacking of coherence between text and table 3. All outcomes under the “Bone health” label are biomarkers of osteoporosis. We re-labelled these outcomes from “Osteoporosis” to “Bone health” in Table 3.

Line 255 ….>50% higher risk…

Thanks for the suggestion, this has been corrected (line 61).

Line 259 not associated with CV…

Thanks for the suggestion, this has been corrected (line 264).

Line 263 …heart diseases, cerebrovascular diseases can hardly be separated from 3.2.2. How did you manage this?

We thank the Reviewer for pointing out this problem, the chapters have been joined in the revision (lines 257-272).

Section 3.2.4 on bone and muscle health does not include any information on muscle health

Thanks for the suggestion. We deleted “muscle health” from the label of the outcome category.

Line 299 …with marker of …?

Thanks for the suggestion, this has been corrected (line 309).

Section 3.2.9 may be shifted to 3.2.4, what about the other secondary outcomes? Were there no results?

We thank the Reviewer for the suggestion, paragraph 2.5 has been now modified to improve clarity. According to the definition of “healthy ageing” adopted for this systematic review, physical functionality dimension includes falls as markers (Table 1).

Line 310 What is meant by physical functionality? It seems to be missing in Table 4

Thank you for making us notice the mistake, it has been now deleted.

Discussion

Large part of the discussion repeats the results.

We have largely modified the discussion according to the Reviewer’s suggestions and hope that, in its current form, has given sufficient space to data interpretation according to current literature.

Line 326 ..between vitamin D and biomarkers?? Mostly health outcomes/diseases?

Thank you for making us notice the mistake, the entire sentence has been now modified.

Line 327 see comment on Line 76

Thanks for the suggestion, this has been corrected (line 346)

Line 334 …as low quality

Thanks for the suggestion, this has now been deleted.

Line 356 ...negative associations were observed with all-cause…?

The sentence has been entirely modified in the revised form of the manuscript

Line 364 …. in young subjects – what does this mean in the context of premature death?

Thanks for the suggestion, this has been now deleted.

Line 371 However, this study focused on cohort with a high proportion of unhealthy participants?

We clarified this point and added the citation (lines 366-377), since this statement was referred to another study and not the present review.

Line 373 Based on the previous evidence,…

Thanks for the suggestion, this has deleted in the current version of the discussion

Lines 376/377 what do you mean by: this is clearly related to the different types of cancer highlighted in studies selected for the current review.

Thanks for the suggestion, this has modified in the current version of the discussion (line 360-361)

Line 392 …on cohorts with healthy younger adults…

Thanks for the suggestion, this has deleted in the current version of the discussion (line 384)

Conclusions

These definite conclusions cannot be drawn from the results of this systematic review. For example, there were only two studies on sarcopenia included and only one with 25(OH)D. Also on pulmonary events, there were different outcomes and one association was only significant in smokers. This section needs to be revised and more cautious conclusions should be drawn from the results.

Thanks for the suggestion, the conclusions were entirely redrafted (lines-402-410).

In general vitamin D instead of vitamin-D, 1,25(OH)D or 1.25(OH)D

Thanks for the comment, all typos and mistakes have been corrected throughout the manuscript

Round 2

Reviewer 1 Report

To correspond with the article title, the conclusion could be rephrased as "vitamin D as multidimensional predictors of ill-health in the ageing process.  Further well-designed controlled trials to investigate whether vitamin D supplement results in superior outcomes are warranted in the future."  

Author Response

To correspond with the article title, the conclusion could be rephrased as "vitamin D as multidimensional predictors of ill-health in the ageing process. Further well-designed controlled trials to investigate whether vitamin D supplement results in superior outcomes are warranted in the future."

We thank the Reviewer for this comment. As advised, the conclusions in the abstract (lines 27-30) and text have been modified accordingly (lines 410-412).

Reviewer 2 Report

The manuscript has improved a lot. However, there are still some confusing sections.

Lines 261-266 and 268f are in duplicate.

Table 3

It doesn’t make sense to report on bone health and show negative associations with vitamin D. All results refer to bone loss and fractures. Thus, I would suggest “Impaired bone health” as a category, otherwise the associations would be positive.

Additionally, it is very confusing that there were 7 negative associations for “bone health” but if you count them in the 3.2.4 section it seems only to be 5.

Table 3, description: “the last column reports non-statistically significant associations between 25(OH)D levels at baseline and risk of an event at follow-up” – As the last column shows the total number of associations, the column entitled “Non differences” may be more correct.

Section 3.2.6, Lines 296f: This description seems not to be correct as it is said that all associations were not significant except for one study reporting a direct association with non-melanoma skin cancer. The direction of this association should be mentioned. The section goes on with another significant association with cancer mortality which also seems to be included in Table 3. Thus, it is confusing why it is said that only one significant association was found.

Line 268: “cardio death”? – I would suggest: cardiovascular and metabolic death

Line 342-343 “and described in detail 342 by Depp et al. [32]”? Does this mean “as described by…”?

Line 343: “Opposed to all these the large…“ Should this read „Opposed to the large majority…”?

Line 350: Be more specific: “…the higher costs of measuring 1,25(OH)2D.”

Line 357: Please revise as you did not find an inverse association of bone health and vitamin D but on bone loss or fractures and vitamin D.

Line 361: colorectal cancer

Lines 398-399:  There is no conclusion on “bone mineral density….cancer events” – these outcomes were just added at the end of the sentence. Please revise this sentence.

Author Response

We thank the Reviewer for the enclosed comments and for pointing out further imperfections and unclear phrasing.

Lines 261-266 and 268f are in duplicate.

We apologize for the mistake, which has been now corrected (line 261-264)

Table 3 - It doesn’t make sense to report on bone health and show negative associations with vitamin D. All results refer to bone loss and fractures. Thus, I would suggest “Impaired bone health” as a category, otherwise the associations would be positive.

As suggested, category name in Table 3 and 3.2.4 section title have been revised.

Additionally, it is very confusing that there were 7 negative associations for “bone health” but if you count them in the 3.2.4 section it seems only to be 5.

We have verified and consequently modified the section as highlighted.

Table 3, description: “the last column reports non-statistically significant associations between 25(OH)D levels at baseline and risk of an event at follow-up” – As the last column shows the total number of associations, the column entitled “Non differences” may be more correct.

Table caption and column heading has been revised both in Tables 3 and 4.

Section 3.2.6, Lines 296: This description seems not to be correct as it is said that all associations were not significant except for one study reporting a direct association with non-melanoma skin cancer. The direction of this association should be mentioned. The section goes on with another significant association with cancer mortality which also seems to be included in Table 3. Thus, it is confusing why it is said that only one significant association was found.

The associations were verified. We apologize for the puzzling phrasing, the section (lines 297-302) and Table 3 have been revised.

Line 268: “cardio death”? – I would suggest: cardiovascular and metabolic death

The abbreviation has been replaced as suggested (line 268).

Line 342-343 “and described in detail 342 by Depp et al. [32]”? Does this mean “as described by…”?

This has been corrected (lines 345-346).

Line 343: “Opposed to all these the large…“ Should this read „Opposed to the large majority…”?

This has been modified as requested (line 346).

Line 350: Be more specific: “…the higher costs of measuring 1,25(OH)2D.”

This has been revised as requested (line 353).

Line 357: Please revise as you did not find an inverse association of bone health and vitamin D but on bone loss or fractures and vitamin D.

This has been revised as requested (line 359).

Line 361: colorectal cancer

This has been modified as requested (line 364).

Lines 398-399:  There is no conclusion on “bone mineral density….cancer events” – these outcomes were just added at the end of the sentence. Please revise this sentence.

To add clarity, the entire introductory sentence of the discussion has been revised (lines 400-405).